# Automated relative binding free energy calculations from SMILES to ΔΔG

J. Harry Moore [1], Christian Margreitter[1], Jon Paul Janet [1], Ola Engkvist [1✉], Bert L. de Groot [2✉] & Vytautas Gapsys [2✉]

In drug discovery, computational methods are a key part of making informed design decisions and prioritising experiments. In particular, optimizing compound affinity is a central concern during the early stages of development. In the last 10 years, alchemical free energy (FE) calculations have transformed our ability to incorporate accurate in silico potency predictions in design decisions, and represent the 'gold standard' for augmenting experiment-driven drug discovery. However, relative FE calculations are complex to set up, require significant expert intervention to prepare the calculation and analyse the results or are provided only as closed-source software, not allowing for fine-grained control over the underlying settings. In this work, we introduce an end-to-end relative FE workflow based on the non-equilibrium switching approach that facilitates calculation of binding free energies starting from SMILES strings. The workflow is implemented using fully modular steps, allowing various components to be exchanged depending on licence availability. We further investigate the dependence of the calculated free energy accuracy on the initial ligand pose generated by various docking algorithms. We show that both commercial and open-source docking engines can be used to generate poses that lead to good correlation of free energies with experimental reference data.

[1] Molecular AI, Discovery Sciences, R&D, AstraZeneca, Gothenburg, Sweden. [2] Computational Biomolecular Dynamics Group, Department of Theoretical and Computational Biophysics, Max Planck Institute for Multidisciplinary Sciences, Am Fassberg 11, D-37077 Göttingen, Germany. ✉email: ola.engkvist@astrazeneca.com; bgroot@gwdg.de; vgapsys@gwdg.de

A longstanding challenge in computational drug discovery has been the development of methods for efficient in silico prioritization of compounds. In recent years, relative binding free energy (RBFE) calculations have become the gold standard for accurately computing binding affinities[1–4], a key property during small molecule optimization. However, these simulations remain complicated to set up, computationally expensive to run and technically challenging to scale to large sets of compounds. In general terms, a standard FE calculation workflow first requires generation of a plausible low-energy conformer for each compound from a set of congeneric binders, ideally with consistent formal charge. The resulting embedded ligands are then docked into a protein crystal structure, typically requiring constraints or hand-modelling to generate acceptable poses similar to that of a known binder. Atoms for perturbation are then mapped between pairs of proximate compounds (based on compound similarity), and a relative change in binding free energy is computed for each edge. Typically, around 60 ns of total simulation time is required for a single perturbation[1,3,4].

Several approaches have yielded accurate predictions for large scale RBFE calculations. Equilibrium free energy perturbation (FEP) calculations, popularised by Schrödinger's FEP+ workflow, employ discrete windows along the alchemical coordinate (usually denoted lambda) for a step-wise morphing from one compound to the other[1,5]. Individual simulations at discrete lambda states (typically 12-24 states depending on ligand similarity and charge state changes) can be run in parallel and employ replica exchange with solute tempering (REST) type enhanced sampling between replicas[6].

An alternative approach, employed throughout this work, is non-equilibrium switching (NES)[3]. In this approach, one equilibrium simulation is run for each physical end state of a perturbation, after which many short non-equilibrium simulations ('transitions') are run during which the lambda value is continuously driven from one endstate to the other, and the free energy change is computed using thermodynamic integration[7]. Whilst the overall simulation time per edge is identical to the equilibrium FEP approach, the transition simulations are independent, and therefore trivially parallelisable into many small jobs, allowing for routine evaluation of many edges simultaneously, even on a typical shared computing cluster.

Whilst the outlined FE calculation methods facilitate scaling to large sets of compounds, the requirement for significant investment of expert time to manually set up, execute and evaluate such calculations remains a limitation to wider adoption, particularly outside of industrial research where access to commercial packages is more limited. To address this, we report the development of a fully automated end-to-end workflow employing constrained embedding and docking steps to facilitate non-equilibrium simulation (NES) based RBFE calculations starting from SMILES string ligand representations. The workflows are implemented using Icolos, our in-house, open-source workflow manager, which streamlines the configuration of complex, multi-step physics-based workflows and allows for flexible combination of commercial and open-source tools at various stages of the workflow[8].

Herein we demonstrate the use of various configurations of our workflow to automate the whole process of RBFE calculation: from the SMILES string for ligand representation to the final binding free energy estimate. We showcase how the steps in the workflow can be performed by both commercial and open-source software. In particular, we investigate the impact of automated docking protocols on the quality of RBFE predictions, comparing the performance of both open-source and commercial docking engines, and further demonstrate the deployment of the workflow on both in-house and cloud compute platforms.

## Results and discussion

Automated RBFE calculations were successfully performed for a total of 1005 alchemical perturbations, comprising 201 perturbations between the 127 ligands across the 4 systems for each of the 5 tested FE workflow configurations.

**Impact of the pose generation method.** Previous work on benchmarking FE methods has predominantly relied on hand-modelled ligand poses. Whilst this has allowed for reliable comparison between different binding free energy methods, the performance observed in these benchmarks cannot be expected to translate to prospective, fully automated applications, since the quality of the poses is highly dependent on the docking protocol.

The impact of ligand pose quality has recently been examined in conjunction with Schrödinger's FEP+ workflow[9] where it was shown that high quality poses were key to obtaining reliable RBFE predictions. In particular, it was shown that a core constraint derived from a maximum common substructure (MCS) relative to a reference pose could reliably generate poses from which FEP+ results strongly correlated with experimental binding affinities, especially for cases where the consistent binding mode assumption holds.

Accurate binding mode prediction is particularly crucial when developing automated workflows for free energy calculations which can be later incorporated into more complex pipelines, e.g. active learning protocols that use alchemical calculations to inform surrogate models[10,11]. For such applications one must rely on high quality docking poses, since manual user inspection is not possible in a large scale automated process. In lead optimization applications, where RBFE calculations see the majority of use, this is normally managed by assuming a constant binding mode. This allows poses to be constrained to match a crystallographically determined binding mode, however induced-fit approaches have recently been shown to deliver further accuracy improvements, particularly where homology models have been used[12].

*P38α.* For P38α, poses were successfully generated for all but one ligand under all tested docking protocols. The core-constrained Glide protocol required setting a fallback core constraint threshold of 1 Å which resulted in several poses where the aryl fluoride motif differed from the modelled poses (Fig. 1). Vanilla Glide predicted the overall ligand position in accord with the modelled ligands, however 4 poses have the core out of plane compared to the reference, resulting in the aryl fluoride position also deviating from the reference. In the MCS Glide docking protocol, the experimental orientation of the aryl fluoride ring was predicted consistently with the modelled poses in all but two cases. The AutoDock Vina MCS workflow generated poses with the orientation overall matching that of the other protocols, but with greater deviations in the core position compared to all Glide protocols. In this case, the filtered Vina protocol performed comparably to the vanilla protocol: relatively few poses were generated by the vanilla docking approach, thus filtering provided comparatively little benefit for this system.

These trends are reflected in the quality of the RBFE correlations with experiment. All but Vanilla Glide poses resulted in a correctly predicted overall correlation (i.e. positive Kendal's $\tau$ and Pearson's $\rho$), with MCS and core-constrained Glide performing best in terms of overall correlation, average unsigned error (AUE) and root mean square error (RMSE). The larger variation in core positioning is reflected in higher overall errors for both Vina docking protocols (RMSE=1.7 kcal mol$^{-1}$ for both Vina protocols c.f. 1.1 kcal mol$^{-1}$ for MCS Glide). For Vanilla Glide, we hypothesize that the inconsistent positioning of the aryl

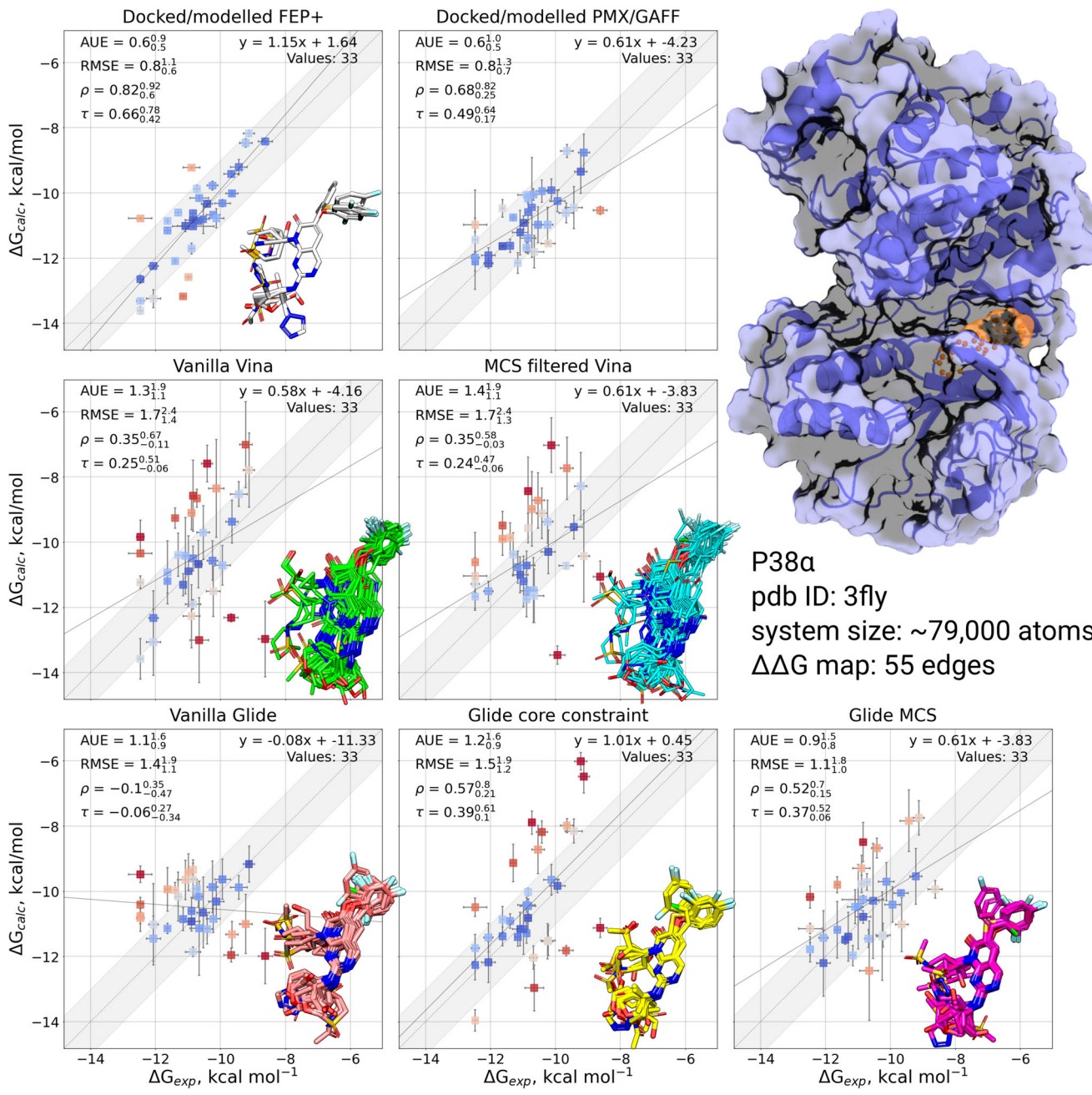

**Fig. 1 Comparison of experimentally measured and calculated binding affinities for P38α protein-ligand system.** The simulations were initialized with different starting structures (docked/modelled or docked only). The data in the panel Docked/modelled FEP+ and Docked/modelled PMX/GAFF are from Gapsys et al.[3]. Overlays of starting ligand poses are inset into the corresponding plot. Experimental errors are taken from the original publication, whilst prediction errors correspond to the standard error over 3 simulation replicas also taking into account the bootstrapped uncertainty of individual free energy estimates. The shaded grey region corresponds to a 1 kcal mol$^{-1}$ deviation from experiment, and the solid line is a linear fit through the data. The crystal structure with a bound ligand highlighted in orange is shown in the top right.

fluoride ring is responsible for the larger errors observed in the RBFE correlations. All automated protocols are outperformed by the manually modelled poses (AUE and RMSE ranged between 0.9-1.4 kcal mol$^{-1}$ and 1.1-1.7 kcal mol$^{-1}$ c.f. 0.6 kcal mol$^{-1}$ and 0.8 kcal mol$^{-1}$ respectively for modelled poses), illustrating the importance of expert knowledge for this system, however only vanilla Glide fails to recover the experimental trend. In contrast, the Glide MCS protocol provided the best performance among the docking protocols with RMSE and AUE values of 1.1 kcal mol$^{-1}$ and 0.9 kcal mol$^{-1}$.

*PTP1B.* Ligands in the PTP1B series contain two carboxylic groups, both of which were modelled in their deprotonated form. As in GAFF2 force field the oxygens of a deprotonated carboxy group were represented by the same atom type carrying identical partial charges, this symmetry removed ambiguity for the initial orientation of the carboxy groups. In addition, the ligands contained a hinge, and as such proved to be a more challenging set to dock, with one ligand failing to redock under all docking protocols due to a steric clash with the receptor. Of the remaining ligands, their orientation in the binding pocket was consistently

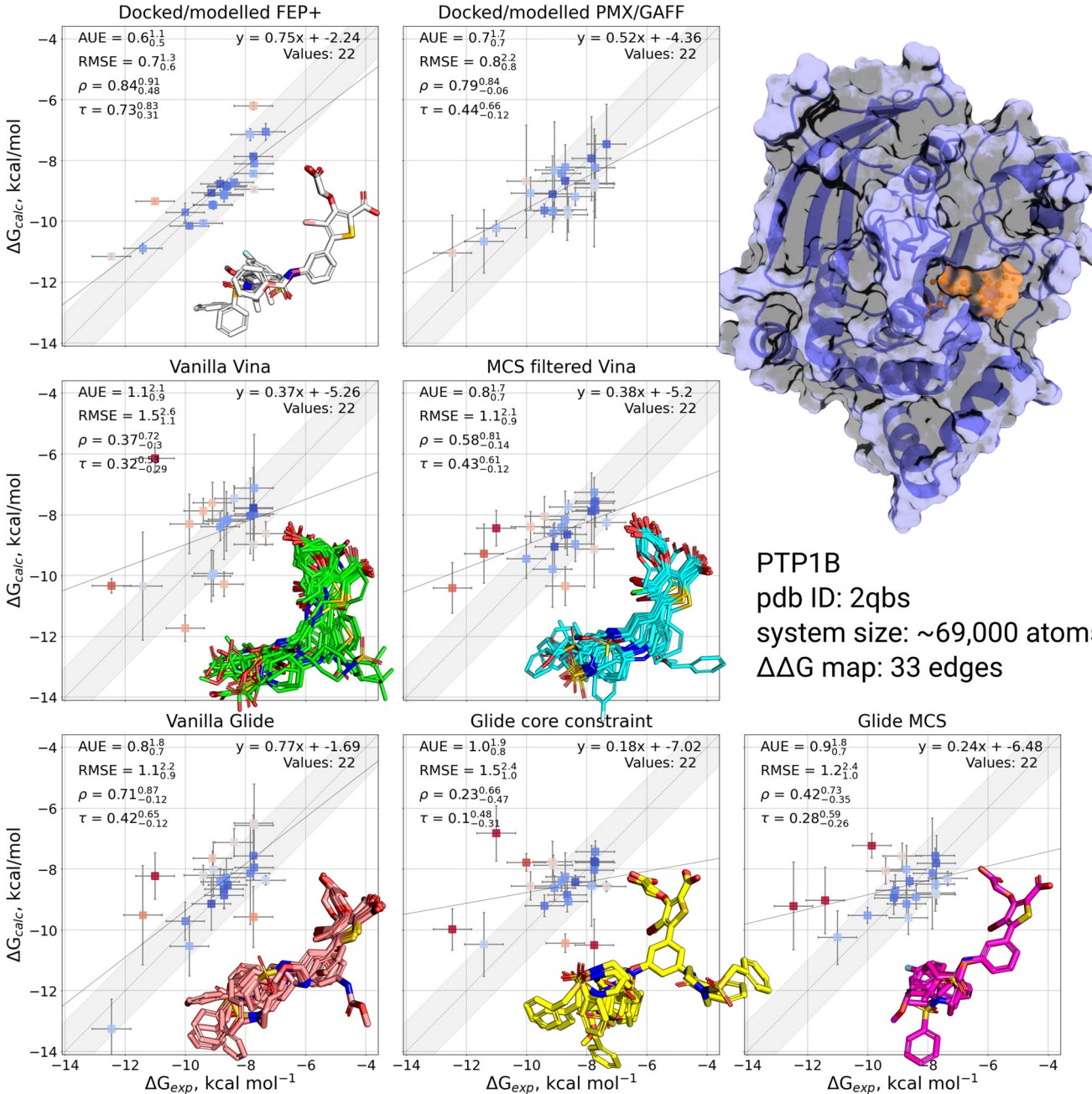

**Fig. 2 Comparison of experimentally measured and calculated binding affinities for PTP1B protein-ligand system.** The simulations were initialized with different starting structures (docked/modelled or docked only). The data in the panel Docked/modelled FEP+ and Docked/modelled PMX/GAFF are from Gapsys et al.[3]. Overlays of starting ligand poses are inset into the corresponding plot. Experimental errors are taken from the original publication, whilst prediction errors correspond to the standard error over 3 simulation replicas also taking into account the bootstrapped uncertainty of individual free energy estimates. The shaded grey region corresponds to a 1 kcal mol$^{-1}$ deviation from experiment, and the solid line is a linear fit through the data. The crystal structure with a bound ligand highlighted in orange is shown in the top right.

predicted differently from the modelled poses in all but the Glide MCS protocols. Similarly to P38α, a fallback core constraint of 1 Å was required to allow the remaining ligands to dock with the fixed core constraint, which resulted in a flipped core being observed for several poses in the Glide core-constrained protocol. Vanilla Glide docking also suffered from a poorly positioned core in some cases (Fig. 2). Vanilla Vina generated a mix of core orientations: in 8 out of 23 ligands the pose was reversed in comparison to the modelled compounds. The MCS-filtered Vina protocol had no such pose inversions, which is reflected in the reduced AUE (1.1 vs 0.8 kcal mol$^{-1}$) and RMSE (1.5 vs 1.1 kcal mol$^{-1}$).

Similarly to P38α, PTP1B's reference poses were extensively hand-modelled in the original publication. This is reflected in the reduced RMSE and increased τ values for the reference computations compared to all tested docking protocols. Nevertheless, all protocols were able to recover the expected trend in experimental binding affinities, with core-constrained Glide producing the worst correlation (Kendal's τ = 0.1). Interestingly, the MCS-filtered Vina together with vanilla Glide protocols achieved the lowest RMSE (0.8 kcal mol$^{-1}$) and AUE (1.1 kcal mol$^{-1}$) among the docking protocols. Also, the worst performing unguided vanilla Vina protocol (RMSE = 1.5, AUE = 1.1 kcal mol$^{-1}$)

performed comparably with the best performing MCS Glide protocol (RMSE = 1.2 kcal mol$^{-1}$, AUE = 0.9 kcal mol$^{-1}$).

*TNKS2*. Poses were successfully generated for all ligands and were, in general, the most consistently generated across the docking protocols of the four systems studied. All methods positioned the core in a comparable manner, with both vanilla protocols successfully identifying the consistent orientation with the other methods, and both MCS and core-constrained protocols producing similar poses by visual inspection.

These docking poses resulted in highly correlated FE predictions, with all methods performing at least on par with, and in some cases better than, the reference poses used in calculations with NES. In particular, MCS Glide produced excellent experimental correlation (Kendal's $\tau$ = 0.67), with AUE = 0.4 kcal mol$^{-1}$ on par with experimental uncertainty (Fig. 3).

The TNKS2 case illustrates well the impact of manual intervention in generating appropriate poses when compared to those used for the P38$\alpha$ and PTP1B cases. Here, the reference results from Schindler et al.[4] and Gapsys et al.[13] studies do not outperform automated docking protocols in terms of agreement with experiment. This effect is likely due to the starting pose generation, where Schindler et al. primarily relied on docking followed by a closer exploration of possible tautomers and multiple binding poses, yet no manual pose generation was attempted.

*SYK*. Poses for all SYK ligands were generated successfully for all docking protocols. Use of both the fixed core constraint and MCS with Glide resulted in consistent ligand positioning (Fig. 4). The MCS protocol was able to further consistently predict the orientation of the phenol ring, whereas some ring flips were observed in the core-constrained protocol. Without a core constraint, the vanilla Glide poses showed larger diversity in core placement, in one case resulting in a flipped ligand. For AutoDock Vina, the vanilla protocol successfully placed the hinge in the orientation matching that of the docked/modelled poses in all cases, whilst the MCS filtering refined the overall alignment, in particular for the phenol ring placement.

Correlations with experimentally measured affinities for SYK were overall comparable to the reference performance, for both FEP and NES. Notably, all workflows meet or improve upon 1 kcal mol$^{-1}$ average AUE, despite a comparatively large, flexible series of ligands compared to other systems under study (Fig. 4). Both Vina protocols achieved similar accuracy to the Glide based docking. All the approaches had difficulties accurately predicting binding affinities at least for some ligands: in each case there are significant outliers both over- and underpredicting binding free energies.

**Overall accuracy and docking protocols**. From the summary of the results across four studied protein-ligand systems (Fig. 5) it is evident that manual modifications of the poses help improving the accuracy of free energy calculations. The poses for P38$\alpha$ and PTP1B were initially generated in[1] and subsequently manually refined in[3]: for these cases the docked/modelled poses outperform the fully automated docking algorithms. For the other two systems, TNKS2 and SYK, the docked/modelled poses did not undergo extensive manual modifications[4] and this is clearly reflected in the accuracy measures, both in terms of correlation with experimental $\Delta\Delta G$ and the ability of the automated workflow introduced here to recover equivalent performance to the original publications (Fig. 5).

For the automated docking protocols, the outcome may strongly vary depending on the system studied, e.g. TNKS2

appears to be a convenient target for the explored set of ligands, while other cases were more challenging. Including core or MCS based constraints into Glide protocol also does not always facilitate accurate prediction of binding affinities. Similarly, MCS filtering for the open source Autodock Vina docking protocol may improve final prediction accuracy, but this effect is also system dependent. Overall, it is encouraging to see that the calculations based on the Vina poses can perform on par with the poses generated with the commercial software, demonstrating the feasibility of a fully open-source end-to-end RBFE workflow.

**How sensitive are the predictions to the starting pose?** From the studied cases it is evident that the outcome of the alchemical free energy calculation will depend on the starting ligand pose. For example, for the TNKS2 system (Fig. 3) prediction accuracy in terms of AUE doubles when changing the starting poses from MCS filtered Vina variant to Glide MCS: $0.8_{0.7}^{1.3}$ and $0.4_{0.4}^{1.0}$ kcal mol$^{-1}$, respectively. Such an accurate prediction, as observed in the latter case, provides confidence in the correctness of the initial ligand poses, which in turn allows establishing a reliable structure activity relationship (SAR) depicted in (Fig. 6).

Here, we show the substituents on a common scaffold together with the highest affinity of a ligand with the corresponding substituent among the investigated congeneric series. The inner circle corresponds to the experimental $\Delta G$, while the outer circle marks the values calculated starting from the Glide MCS based docked ligands. Naturally, such a depiction is ignorant of the relationships between the modified ligand sites, and only provides an insight whether a particular modification of a specific site can lead to high/low affinity. Even such a simplified representation can give a quick intuitive understanding. For example, bulkier hydrophobic modifications at the R4 site lead to higher binding affinity and R5 site has a clear preference for carbon to nitrogen. At the R2 and R3 sites hydrogen and fluorine lead to higher affinities, while at R1, hydrogen and methyl would be preferred.

Of course, in the current study such analysis is only retrospective, as the binding affinities are readily available. Since for this illustration we have selected a well behaved case (affinities in the inner and outer circles of Fig. 6 match well), the constructed SAR provides a reliable interpretation for ligand activity. Yet, if the free energies calculated with some ligand poses were to lead to a disagreement with experiment, obtaining a reliable SAR would not be feasible. For example, Vina based docking as well as docked/modelled poses for the ligand with the nitrogen atom at the R5 position placed the atom on the opposite side than depicted in Fig. 6 (i.e. pointing in the same direction as R3 site). In turn, such poses lead to inaccurate affinity predictions for these cases. For a prospective study, we would need to rely on the calculated affinities based on the docked poses only, i.e. without an experimental binding affinity reference.

There are numerous further complications that might arise when applying such a docking based approach to calculate binding affinities prospectively. One of such difficult situations is depicted in Fig. 7. In this case only the vanilla Vina docking pose yields an accurate free energy estimate ($-9.4 \pm 0.7$ kcal mol$^{-1}$ c.f. $-9.7 \pm 0.2$ kcal mol$^{-1}$ for experiment), however, this starting structure is reversed in comparison to the pose predictions by all other docking protocols. This suggests that consistency in pose prediction among docking algorithms does not necessarily warrant accurate affinity estimation.

Another example of this effect is the vanilla Vina protocol applied to the PTP1B system: more than 33% of the PTP1B vanilla Vina poses were reversed in comparison to other approaches (Fig. 2). Although, starting simulations with such poses deteriorates prediction accuracy, even in this case the

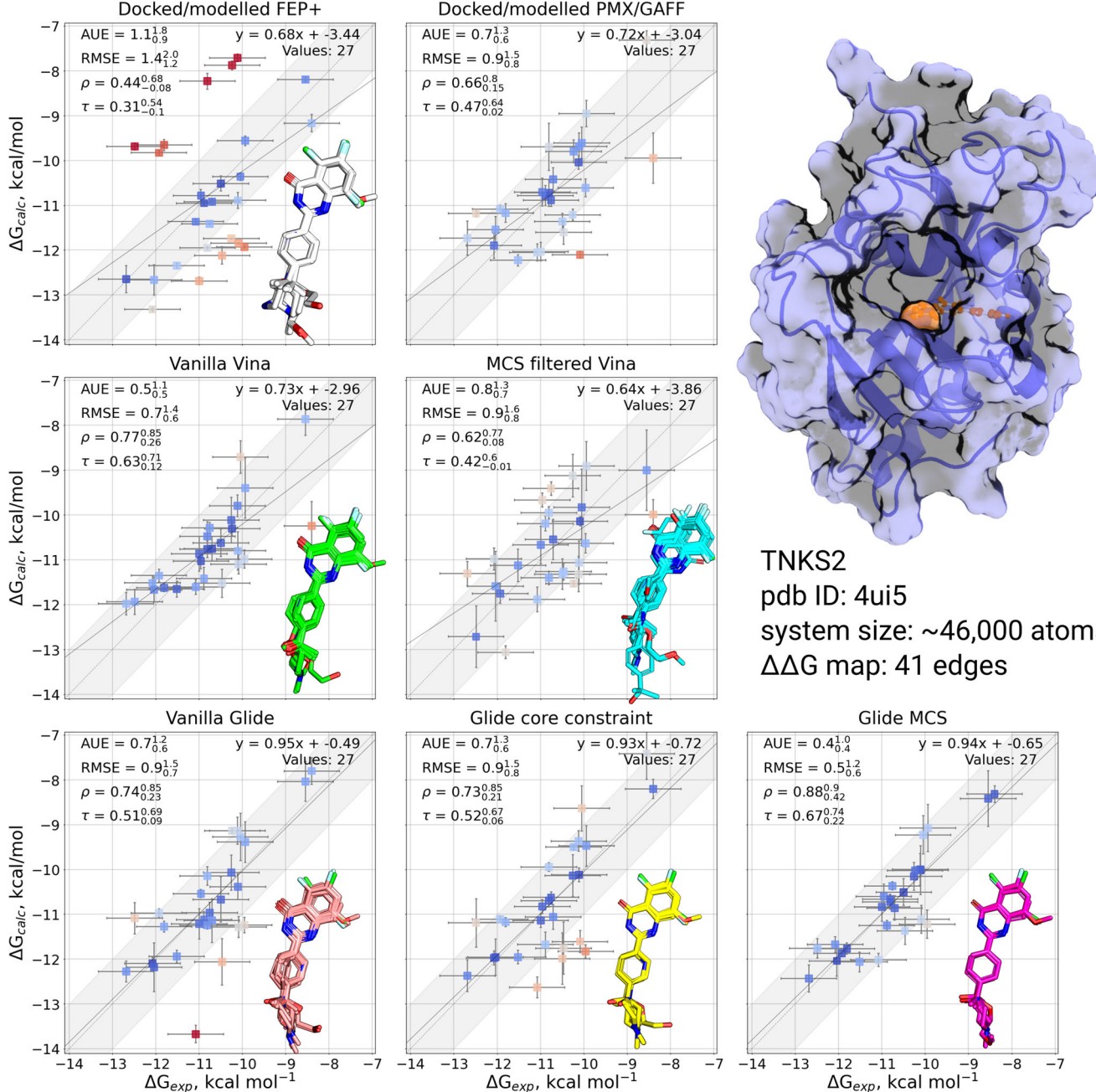

**Fig. 3 Comparison of experimentally measured and calculated binding affinities for TNKS2 protein-ligand system.** The simulations were initialized with different starting structures (docked/modelled or docked only). The data in the panel Docked/modelled FEP+ are from Schindler et al.[4]. The data in the panel Docked/modelled PMX/GAFF are from Gapsys et al.[13]. Overlays of starting ligand poses are inset into the corresponding plot. Experimental errors are taken from the original publication, whilst prediction errors correspond to the standard error over 3 simulation replicas also taking into account the bootstrapped uncertainty of individual free energy estimates. The shaded grey region corresponds to a 1 kcal mol$^{-1}$ deviation from experiment, and the solid line is a linear fit through the data. The crystal structure with a bound ligand highlighted in orange is shown in the top right.

overall ΔG accuracy is not significantly worse than for the rest of the docking protocols (Fig. 5). These examples also highlight the sometimes unexpected insensitivity of the binding affinity predictions to the starting pose.

Another caveat to take into account when correlating docking pose quality with alchemical calculation accuracy is that the reported absolute ΔG values are only reconstructions from the relative ΔΔG estimates[14]. This means that accurate/inaccurate predictions may not necessarily be caused by the inappropriate pose of a ligand itself, but also of its neighbours to which it was mapped in the ligand perturbation map. All in all, for a systematic

study of the starting pose effects on the affinity prediction accuracy one would require a particular system setup with well controlled structural changes or rely on absolute binding affinity calculations[15–17].

**Deploying the workflow in the Cloud.** To demonstrate the potential of our end-to-end RBFE workflow, we benchmarked it using the Amazon Web Services (AWS) cloud. For this proof of concept demonstration we calculated relative binding free energy between two ligands from the set of TNKS2 inhibitors. By

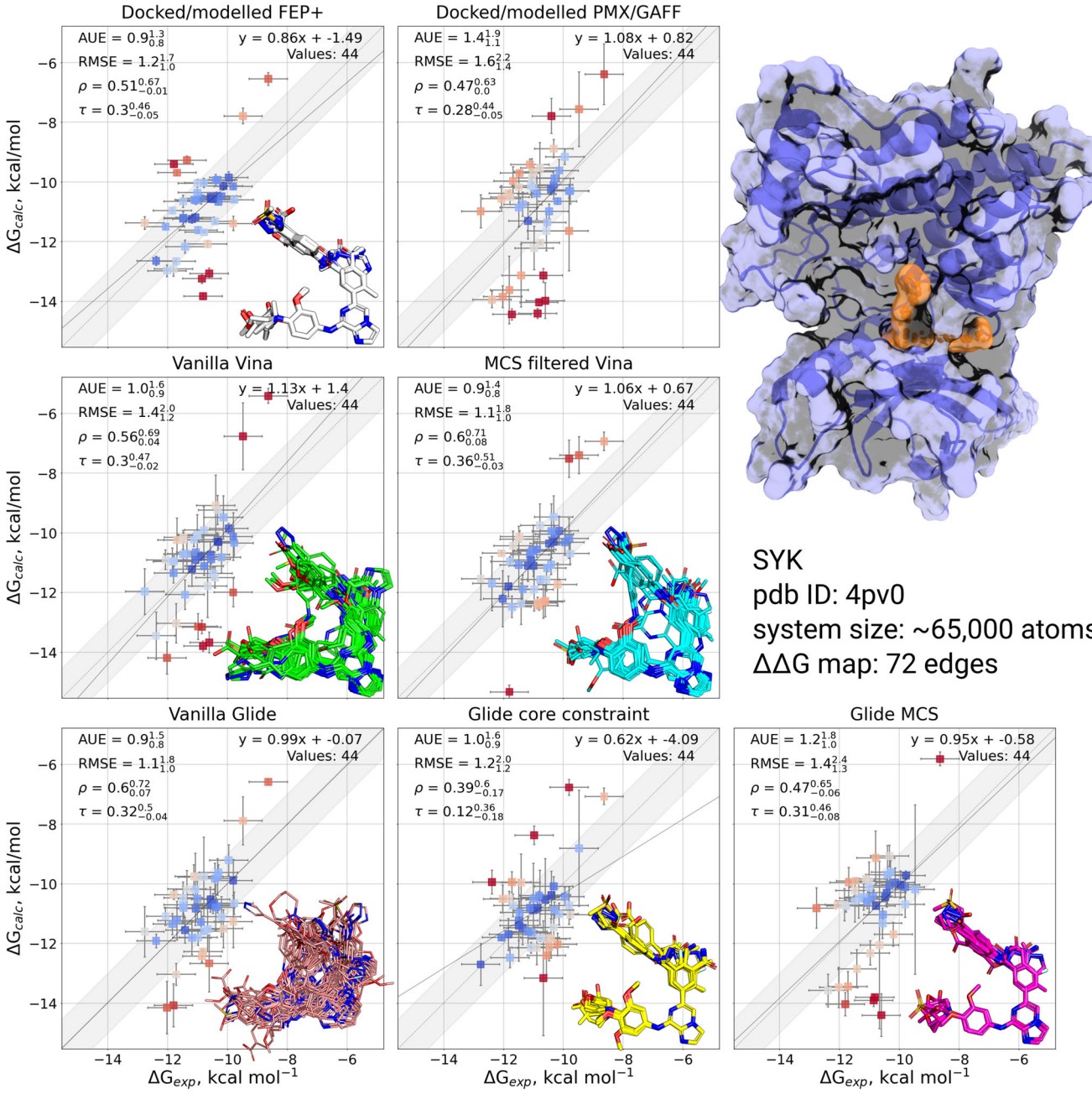

**Fig. 4 Comparison of experimentally measured and calculated binding affinities for SYK protein-ligand system.** The simulations were initialized with different starting structures (docked/modelled or docked only). The data in the panel Docked/modelled FEP+ are from Schindler et al.[4]. The data in the panel Docked/modelled PMX/GAFF are from Gapsys et al.[13]. Overlays of starting ligand poses are inset into the corresponding plot. Experimental errors are taken from the original publication, whilst prediction errors correspond to the standard error over 3 simulation replicas also taking into account the bootstrapped uncertainty of individual free energy estimates. The shaded grey region corresponds to a 1 kcal mol$^{-1}$ deviation from experiment, and the solid line is a linear fit through the data. The crystal structure with a bound ligand highlighted in orange is shown in the top right.

leveraging an AWS ParallelCluster and Icolos' SLURM interface, each simulation type was dispatched to a different instance to showcase the versatility of the approach in maximising the price-to-performance ratio. Optimal node selection for running GROMACS workflows in the cloud has been recently studied[18]. Based on this work, minimisation and NVT equilibration jobs were performed using Intel CPU-only C5 instances, equilibrium simulations using NVIDIA-equipped G4DN nodes, and transitions using AMD HPC6a. All in all, with our Icolos workflow, a single free energy estimate requires 10 hours of wall clock time, at a cost of $12-15 per $\Delta\Delta G$ value.

To support this running mode and improve the scalability of the workflow, we integrated a fault-tolerant capability to enable the use of spot-allocated instances on AWS. By efficiently handling revoked node allocations, and restarting interrupted simulation from a checkpoint file as soon as another node is available, price/performance is significantly improved.

## Conclusion

In this work, we have introduced a framework for end-to-end calculation of relative binding free energies using our previously

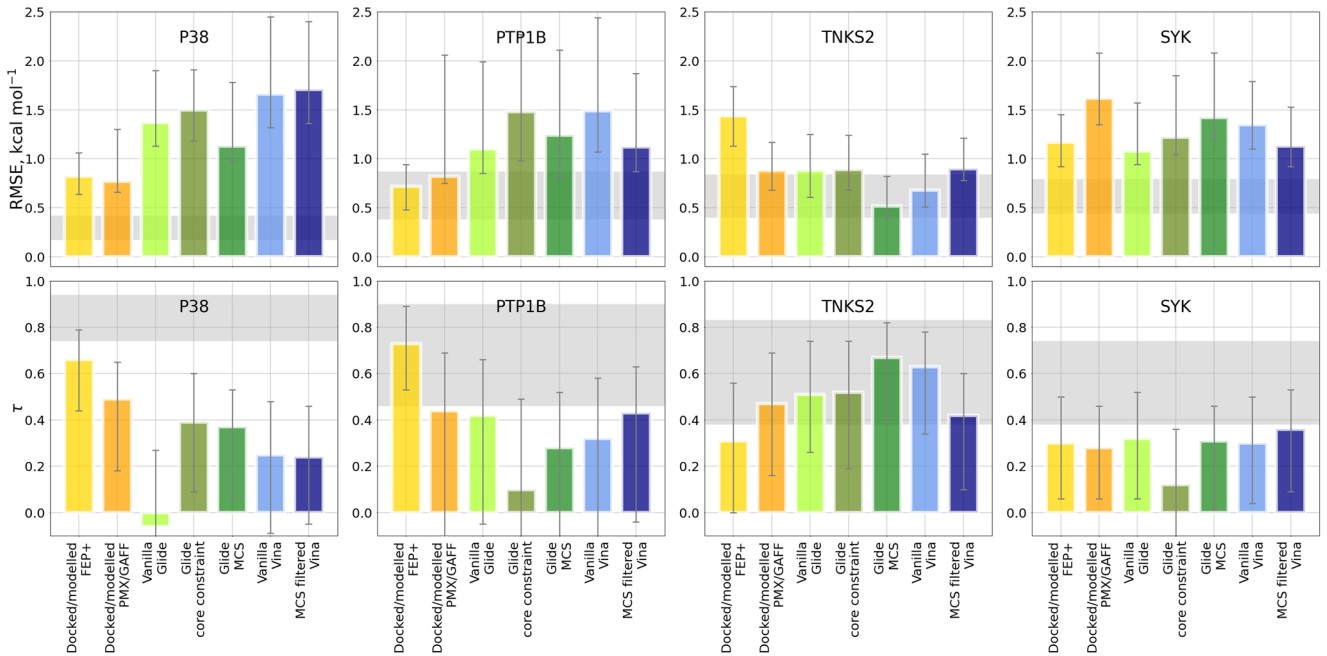

**Fig. 5 RMSE and Kendall's $\tau$ for each docking method on each of the four tested systems.** Shaded regions represent 95% confidence intervals for RMSE and $\tau$ of the experimentally measured values based on the experimental uncertainty. For the cases where experimental uncertainty was not available (PTP1B, TNKS2, SYK), a lower bound for uncertainty of 0.43 kcal mol$^{-1}$ as suggested by the best practices was used[47]. Error bars denote 95% confidence intervals for the calculated values.

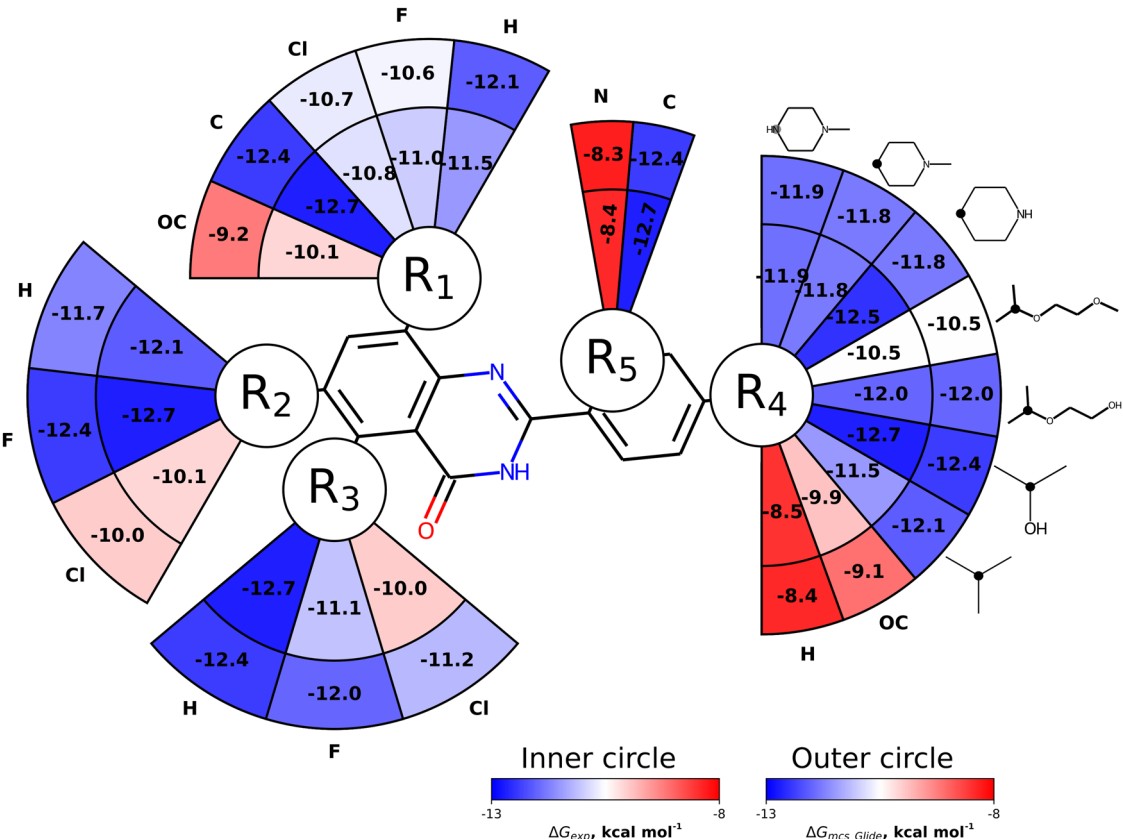

**Fig. 6 Structure activity relationship (SAR) for the ligand series binding to TNKS2.** The affinities of the strongest binder with the corresponding substituent at a given site are shown. Experimental $\Delta G$ is depicted in the inner circle. Calculation based on the poses from Glide MCS are shown in the outer circle.

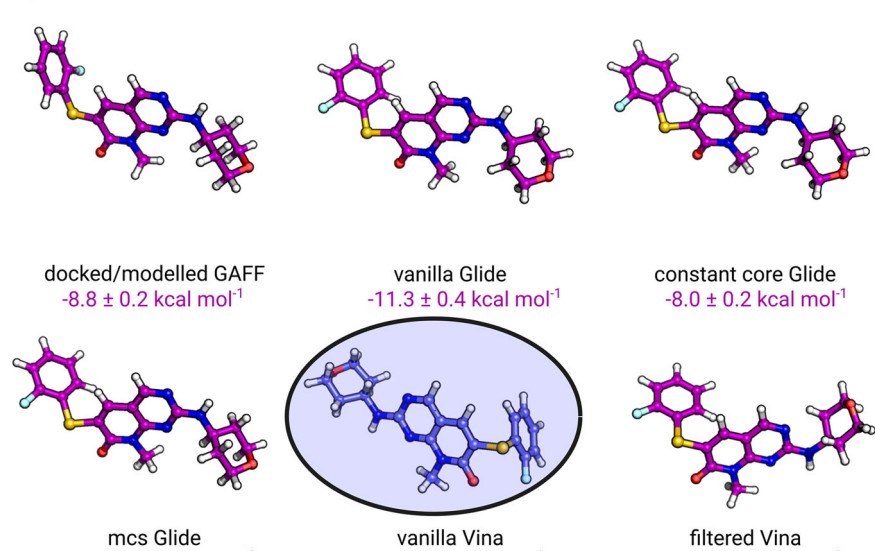

**Fig. 7 Impact of core positioning on ΔG prediction for the ligand 2h in the P38α system.** The vanilla Vina pose delivers an accurate free energy prediction compared to the experimental value of −9.7 ± 0.2 kcal mol$^{-1}$, despite the inverted core (circled) compared to the modelled pose.

developed Icolos workflow manager. This implementation provides a scalable, flexible and reproducible method of executing free energy calculations and allows for straightforward distribution, parallelisation and monitoring of calculations using either local or cloud computing resources. The modular nature of the workflow enables substitution of the blocks allowing to easily combine open source and commercial software packages.

We have employed our workflow to investigate the impact of the docking protocol on automated SMILES-to-ΔΔG calculations, and show that, in general, open-source docking protocols performed competitively with commercial alternatives. Additional constraints on the ligand during docking procedure may improve correlation of the estimated binding affinities with experiment, however this appears to be strongly dependent on the system. Nevertheless, the vast majority of tested configurations, including all core- or MCS-constrained protocols, successfully produced qualitative agreement with experimental binding affinities, demonstrating the value of our automated approach in early-stage drug discovery applications, where the workflow's ability to enrich a compound series by differentiating the most and least potent binders is critical. In this aspect, we have shown that our workflow can, even in challenging cases, produce results that are qualitatively in agreement with those from expert-optimised poses, whilst in more favourable cases provides quantitative agreement to within experimental uncertainty.

The accuracy of binding free energy predictions depends on the quality of docked poses. An experimentally resolved protein structure with a co-crystallized molecule sharing a similar scaffold to the ligands of interest may facilitate higher quality starting pose generation via constrained docking algorithms. However, as demonstrated by several examples in the current work, such constrained docking does not automatically guarantee agreement with experiment in terms of predicted binding affinity. To improve on the accuracy of alchemical free energy calculations numerous other aspects need to be taken into account, e.g. protein and small molecule force field, initial protein and ligand structure preparation, ensuring sufficient sampling, experimental measurement quality.

We hope that, through freely available source-code and example workflow configurations, we have lowered the barrier to entry, and increased the usability of non-equilibrium binding free

energy calculations in drug-discovery applications. Furthermore, as the computational cost of free energy calculations continues to drop with ongoing advances in software and hardware, we believe that automated workflows that retain a high degree of control for the expert user will become increasingly important, as these predictions become part of large computations, for example active learning and *de-novo* design tasks.

## Methods

**Automated SMILES to ΔΔG workflow**. All workflows were implemented using the Icolos workflow manager[8]. Introduction into starting using Icolos is provided in the Supplementary Note 1. Workflows were specified as a series of sequentially executed steps using JSON configuration files. The workflow recipes and full workflow configurations can be found in the Supplementary Note 2 and Supplementary Note 3, respectively. This approach makes the setup straightforward, reproducible and transferable between different protein-ligand systems with only minimal changes required, whilst allowing for efficient parallelisation and utilization of high performance computing (HPC) resources through the built-in Slurm interface. Icolos also enables error checking and failure handling for each job required to evaluate a typical ligand perturbation map.

The steps for a typical end-to-end workflow are shown in Fig. 8, with multiple compatible backends listed where appropriate. Workflows were started with a ligand embedding step, in which a single low-energy conformer for each ligand was generated. This was either performed with LigPrep[19] or RDKit[20]. For LigPrep, default settings were used unless otherwise specified; with EPIK[21,22] calculation was performed at pH = 7 ± 2, enumerating over undefined stereocenters and tautomers and filtering to a single net charge. For RDKit, ligands were embedded and 3D coordinates optimised using the UFF force field[23].

Embedded ligands were subsequently docked into the receptor. For Glide workflows, calculations were performed using Schrödinger version 2021-4, with default settings used throughout, unless otherwise noted. A single conformer was generated for each ligand in all cases[24–27]. In addition to the standard docking protocols, termed 'Vanilla' in this work, we investigated both fixed and maximum common substructure (MCS)-derived positional constraints available in Glide. Whilst equivalent functionality is not available in Vina, we implemented a *post-hoc* filtering step using RDKit functionality to identify the pose for each ligand that minimised RMSD compared to the reference pose[20]. This is implemented in Icolos using the data_manipulation step (see example configuration files for details).

Docked poses for the open-source variants of the workflow were generated using AutoDock Vina 1.2.1[28,29]. For the 'Vanilla' protocol, a single conformer was generated for each ligand, whilst for the MCS filtered protocol, a maximum of 64 conformations per compound were generated, and subsequently filtered by selecting the pose that minimised the RMSD of the maximum common substructure of the ligand compared to a reference ligand.

For each system, a single pose from the reference dataset was selected, from which all positional constraints were derived. Fixed core constraints were manually

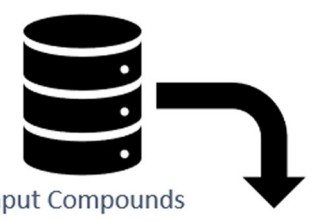

1. Ligand embedding (Ligprep, RDKit)
2. Ligand Docking (Glide, ADV)
3. Optional MCS filtering (ADV)
4. Identify optimal perturbations (Schrodinger, LOMAP)

5. pmx setup
6. pmx atomMapping
7. pmx ligandHybrid
8. pmx assembleSystems
9. pmx boxWaterIons

10. System equilibration (minimisation, restrained NVT)

11. End-state simulations (NPT)

12. Extract snapshots
13. Transition simulations (TI)

14. Analyse TI data, compute ΔΔG

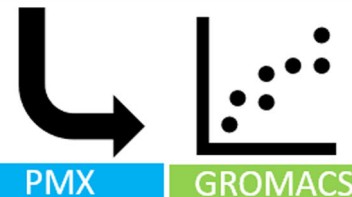

PMX    GROMACS

**Fig. 8 Overview of the steps comprising our automated NES RBFE workflow.** The workflow includes ligand embedding from SMILES, docking, topology preparation, simulation and analysis steps.

specified by SMARTS string to capture the conserved portion of the molecule. Full Glide configuration files for each system are provided in the Supplementary Data.

In total, for each test system, the following five sets of docking poses were generated.

- Glide docking using no constraints ('Vanilla Glide')
- Glide core constraint
- Glide MCS
- AutoDock Vina with no constraints ('Vanilla Vina')
- AutoDock Vina filtered MCS

For the subsequent PMX calculations, multiple bespoke workflow steps were created to prepare, execute and analyse the simulations. In general, individual Icolos PMX steps correspond to the execution of an underlying executable from the PMX package, while further bespoke code was written to handle simulation preparation and execution.

From the docked ligands, a perturbation map was constructed to identify suitable edges based on maximum common substructure matching. This functionality is available in Icolos using either Schrödinger's `fep_mapper.py` script[5] or an open source LOMAP tool[30]. In this work, a single perturbation map was generated for each test system using Schrödinger's `fep_mapper.py` script and the Glide MCS poses to eliminate the effect of changes to the perturbation map topology on the accuracy of RBFE predictions.

Output from the perturbation mapping steps was parsed internally to a unified `PerturbationMap` data structure, capturing the node identity and graph connectivity, and subsequently driving the remainder of the workflow. The PMX calculation was then set up, involving construction of the output directory structure and generation of protein and ligand parameters. The protein was parameterised with `gmx pdb2gmx` using the AMBERff99sb*-ILDN forcefield[31–33]. GROMACS version 2021.6 was used for all preprocessing and simulations. ACPYPE version 2022.1.3[34] and Antechamber version 21.0[35] programs were used to assign GAFF2 parameters and AM1-BCC partial charges[36] for the ligands. `pmx atomMapper` was then run to establish a mapping between the atoms to be morphed between ligand pairs, followed by `pmx ligandHybrid` to generate the GROMACS topologies for hybrid molecules. Full systems were then assembled, and subsequently prepared for simulation using the step `pmx box water ions`. In this step, the standard GROMACS[37] tools were used to place each system in a dodecahedral box with a buffer of at least 1.2 nm from solute to box edge, solvate with the TIP3P water model[38] and neutralise with sodium and chloride ions[39] at a concentration of 0.15 M.

The edges between compounds from the ligand perturbation map were then taken through each step of the simulation protocol, each of which comprised a `prepare_simulation` stage to generate the `tpr` files, followed by a `run_simulation` stage to submit the `gmx mdrun` jobs. First, systems were subjected to a short minimisation job using the steepest descent algorithm, followed by a 10 ps equilibration in the NVT ensemble. Equilibrium simulations were then run for 6 ns in the NPT ensemble, after which transitions were prepared with a bespoke step to handle frame extraction and preparation. The first 2 ns of trajectory were discarded as additional equilibration time, and 80 equally spaced snapshots were then extracted from the last 4 ns of production trajectories, for both the ligand and complex simulations. Each extracted frame served as the start for performing non-equilibrium transitions. 80 such transitions were performed in each direction over 50 ps, and the work done was computed using thermodynamic integration. Finally, the analysis step was run to compute ΔΔG values and errors for each edge using maximum likelihood estimator[40] based on Crooks Fluctuation Theorem[41], as implemented in the `pmx analyse` program, and generate summary files.

**Simulation details**. We use the simulation protocol described in detail by Gapsys et al.[3]. A 2 fs timestep was used throughout. For all simulations following minimisation, the stochastic dynamics integrator was used keeping the temperature at 298 K with an inverse friction of 2 ps. Pressure was held at 1 bar using the Parinello-Rahman barostat[42] with a time constant of 5 ps. Long range coulomb interactions were treated using Particle Mesh Ewald (PME)[43,44] with a real space cutoff of 1.1 nm, using a relative strength at the cutoff of 1e–5. Fourier grid spacing was set to 0.12 nm, and van der Waals interactions were smoothly switched off between 1.0 and 1.1 nm, with dispersion correction for both energy and pressure applied. For bonds to hydrogen atoms, bond lengths were constrained using the LINCS algorithm[45]. For alchemical transitions the van der Waals and electrostatic interactions were soft-cored[46].

**Target selection**. Systems were selected to facilitate comparison with RBFE calculations reported in previous studies,[1,3,4,13] whilst ensuring ligand sets represented a sufficiently diverse set of compounds, including flexible examples that might prove to be challenging for docking algorithms.

Four systems were selected for our investigation, and are summarised in Table 1: P38α and PTP1B were taken from the work of Wang et al.[1]. The ligand poses were further adjusted by Gapsys et al.[3], and are generally considered to be of high quality. Additionally, we selected TNKS2 and SYK from the work of Schindler et al.[4]. The original ligand poses from this study were generated primarily by core-constrained docking using Glide, however, in the case of multiple poses, charge

**Table 1 Summary of systems studied in this work.**

| System Name | Number of Ligands | Exp. ΔG range | Reference Ligand |
|---|---|---|---|
| P38α | 33 | 4.0 kcal mol$^{-1}$ | lig_p38a_2l |
| PTP1B | 23 | 5.0 kcal mol$^{-1}$ | lig_23466 |
| TNKS2 | 27 | 4.3 kcal mol$^{-1}$ | 5n |
| SYK | 44 | 4.5 kcal mol$^{-1}$ | CHEMBL3265037 |

states, or tautomers, all plausible combinations were calculated by FEP+, and the pose with the lowest predicted binding free energy was reported as the input structure. All input data used in this work is available in the Supplementary Data.

**Comparison to experiment**. For comparison to experiment, we converted double free energy differences (ΔΔG) to absolute binding free energies (ΔG). For that we relied on the cycle closure correction procedure and used an average of the experimental measurement for the corresponding congeneric series to offset calculated values.[14] Throughout the work we compared computed free energy values to the experimental measurements in terms of average unsigned error (AUE), root mean square error (RMSE), Pearson's correlation coefficient ($\rho$) and Kendall rank correlation coefficient ($\tau$).

## Data availability

Docked poses, workflow configuration files and calculated free energy values are available at https://github.com/deGrootLab/icolos_pmx_paper_2022.

## Code availability

Icolos (Apache-2.0 license) workflow manager is available at https://github.com/MolecularAI/Icolos. Free energy calculation setup was performed with pmx (LGPL-3.0 license): https://github.com/deGrootLab/pmx.

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

## Acknowledgements
J.H.M. would like to thank Martin Packer for fruitful discussions and feedback during prototyping. V.G. was supported by the BioExcel CoE (www.bioexcel.eu), a project funded by the European Union Contract H2020-INFRAEDI-02-2018-823830.

## Author contributions
J.H.M. and C.M. developed the code, with input from V.G., V.G. and J.H.M. performed calculations and analysed the data. J.H.M. and V.G. wrote the manuscript, which was subsequently revised by all authors. C.M., J.P.J., O.E. and B,L.d.G. supervised the work. All authors approved the final manuscript.

## Funding

## Competing interests
The authors declare no competing interests.

## Additional information

**Peer review information** : *Communications Chemistry* thanks the anonymous reviewers for their contribution to the peer review of this work. Peer reviewer reports are available.

