## [Peer Review File · Communications Chemistry]

Reviewers' comments:

Reviewer #1 (Remarks to the Author):

Free-energy perturbation is a highly powerful computational method that can be employed to estimate the free energy differences between analogs in a drug discovery setup. This method is essential during the lead optimization phase. However, this method is extremely slow and computationally expensive. Therefore, computational and medicinal chemists must provide a rationale for selecting only a few compounds for this analysis. In this paper, the authors generated an end-to-end automated workflow starting from SMILES strings for FEP calculations, employing commercial and open-source tools. The authors provided highly profound and detailed supplementary materials as well as made the code and the data publicly available.

This is a high-quality manuscript and I recommend the manuscript be accepted for publication.

I have minor suggestions:

The authors only discuss the limitations of the tool related to the poses. I believe the quality of this manuscript would improve enormously if the authors would better discuss, providing a system and SAR analysis, the reasons that different systems performed differently. This is particularly important because FEP is a tool for leading optimization. In addition, the authors should highlight the limitations of their tool in the discussion as well as in the conclusions. These would allow users to access their results better and not improperly interpret them.

Reviewer #2 (Remarks to the Author):

The authors present an end-to-end workflow for setting up and performing relative free energy calculations. This is based on earlier work by Gapsys and Groot on non-equilibrium switching and is performed in the pmx package. What is new is that they apply their method to four data sets and present a workflow to get from Smiles string of ligands to the relative binding free energy calculations.

Setting up and performing free energy calculations involves a lot of manual work, such as finding good starting geometries and correct atom mapping.

The authors use four datasets from previous FEP benchmarks (including protein targets p38a, Syk, TNKS2, PTP1B) and show the performance of their workflow as well as the impact of ligand preparation and alignment/docking on the accuracy of FE calculations. They also share some insights on how to set up and run such a workflow system in the cloud.

The publication is relevant to the field and will help researchers set up and run FE calculations semi-automatically. It was a pleasant experience reading the paper.

The authors publish the docking poses and docking configuration files - which is good. Also, the statistical analysis, predicted/experimental dG plots, and binding modes presented are very helpful for understanding. It is not clear from the text and supplementary material whether the code "Optional MCS filtering" has been published. I would argue for publication of the paper if the authors released the code for ligand docking and alignment using MCS (all steps in Figure 1 - I was not sure if steps 1.-4. are fully published).

Suggestions for improvement and questions:

*

Do I you need a FEP+ license to run the fep_mapper.py script?

*

Definition of statistical parameters like AUE is missing.

*

Figure 2: MCS filtered Vina, why is there so much core flexibility/ when MCS was applied?

*

Figure 3: There are two acid functions in the molecule - is it guaranteed that the location of the charge in the superposition is always the same?

*

Figure 7 shows different docking poses of TNKS2 inhibitors. Could you discuss which pose (e.g., pyridine pointing to the NH or CH atom) makes more sense energetically and is consistent with the observed SAR.

Reviewer 1 Comments

I have minor suggestions:

1. **The authors only discuss the limitations of the tool related to the poses. I believe the quality of this manuscript would improve enormously if the authors would better discuss, providing a system and SAR analysis, the reasons that different systems performed differently. This is particularly important because FEP is a tool for leading optimization.**

This is indeed a good suggestion to bring an example of an SAR analysis into the manuscript. For that we have selected a TNKS2 system where accurate affinity predictions were obtained. Furthermore, this nicely illustrates that without a reliable affinity prediction (for which robust ligand pose generation is necessary) such SAR investigation would not be possible.

Exploring the underlying reasons for the different system behavior already goes beyond the scope that we intended for this work. There are numerous underlying issues relating to the force field, protein structure resolution, choices in system preparation, specifics of particular docking algorithms, experimental measurement quality and eventually unknown-unknowns that all will lead to different accuracy observed for different protein-ligand systems. Exploring all these effects would be a very interesting investigation, yet this would also be a different study altogether.

Change in text

Figure 7 and text in the section 'How sensitive are the predictions to the starting pose?'

2. **In addition, the authors should highlight the limitations of their tool in the discussion as well as in the conclusions. These would allow users to access their results better and not improperly interpret them.**

That is a good suggestion: in the restructured Results/Discussion section *How sensitive are the predictions to the starting pose?* we bring up examples of limitations of the approach. In addition, we also updated the Conclusions part with the limitations of the method.

Change in text

Conclusions: The accuracy of binding free energy predictions depends on the quality of docked poses. An experimentally resolved of protein structure with a co-crystallized molecule sharing a similar scaffold to the ligands of interest may facilitate higher quality starting pose generation via constrained docking algorithms. However, as demonstrated by several examples in the current work, such constrained docking this does not automatically guarantee agreement with experiment in terms of predicted binding

affinity. To improve on the accuracy of alchemical free energy calculations numerous other aspects need to be taken into account, e.g. protein and small molecule force field, initial protein and ligand structure preparation, ensuring sufficient sampling, experimental measurement quality.

Reviewer 2 Comments

1. **It is not clear from the text and supplementary material whether the code "Optional MCS filtering" has been published.**

We note that all code, input and configuration files, including all steps referenced in Figure 1, are publicly available through the SI and github repositories for Icolos and the current manuscript:

<https://github.com/MolecularAI/Icolos>

https://github.com/deGrootLab/icolos_pmx_paper_2022

The implementation of MCS filtering is included in the publicly available Icolos source code, and an example workflow configuration using this functionality is also available in the examples folder. We have adjusted the wording in the methods section of the manuscript to clarify this.

Change in text

This is implemented in Icolos using the `data_manipulation` step (see example configuration files for details).

2. **Do I need a FEP+ license to run the `fep_mapper.py` script?**

A license would be required to run the `fep_mapper.py` script, however Icolos supports LOMAP as an open-source alternative for map construction. We have now emphasized this in the text.

Change in text

This functionality is available in Icolos using either Schrödinger's `fep_mapper.py` script [5] or an open source LOMAP tool [20].

3. **Definition of statistical parameters like AUE is missing.**

We have added a definition of the measures used to the methods section.

Change in text

Comparison to experiment

For comparison to experiment, we converted double free energy differences ($\Delta\Delta G$) to absolute binding free energies (ΔG). For that we relied on the cycle closure correction procedure and used an average of the experimental measurement for the corresponding congeneric series to offset calculated

values. [38] Throughout the work we compared computed free energy values to the experimental measurements in terms of average unsigned error (AUE), root mean square error (RMSE), Pearson’s correlation coefficient (ρ) and Kendall rank correlation coefficient (τ).

4. **Figure 2: MCS filtered Vina, why is there so much core flexibility/ when MCS was applied?**

This is certainly a limitation of the *post-hoc* filtering approach, in that it relies on sufficiently close poses being proposed by the docking engine in the initial unguided run. In this case, Vina was unable to generate a sufficiently diverse set of poses to allow MCS filtering to noticeably improve the quality of the core position compared to the standard protocol. We added this explanation to the corresponding section in the text.

Change in text

In this case, the filtered Vina protocol performed comparably to the vanilla protocol: relatively few poses were generated by the vanilla docking approach, thus filtering provided comparatively little benefit for this system.

5. **Figure 3: There are two acid functions in the molecule - is it guaranteed that the location of the charge in the superposition is always the same?**

In the current work initial ligand preparations were generated with Lig-Prep with filter criteria to match the overall charge state of the reference structure. This results in both acids in their unprotonated form where both oxygen atoms in GAFF2 force field are represented by the same atom type carrying identical partial charges. Thus, for the PTP1b case in Figure 3, the location of charges will be preserved and will depend only on the overall ligand positioning by the docking protocol.

Change in text

Ligands in the PTP1B series contain two carboxylic groups, both of which were modelled in their deprotonated form. As in GAFF2 force field the oxygens of a deprotonated carboxy group were represented by the same atom type carrying identical partial charges, this symmetry removed ambiguity for the initial orientation of the carboxy groups.

6. **Figure 7 shows different docking poses of TNKS2 inhibitors. Could you discuss which pose (e.g., pyridine pointing to the NH or CH atom) makes more sense energetically and is consistent with the observed SAR.**

We have now considerably expanded and altered this particular case. Following also the Reviewer 1 suggestion of building an SAR model, we created a much more detailed Figure 7 depicting all the substituents at the

modified sites for this congeneric ligand series. The corresponding text was adapted to focus more on the SAR and the benefits, as well as potential issues when interpreting ligand activity based on calculated affinities that inherently carry dependence on the initial ligand poses.

Change in text

Figure 7 and text in the section 'How sensitive are the predictions to the starting pose?'

REVIEWERS' COMMENTS:

Reviewer #1 (Remarks to the Author):

The authors have addressed all my comments and I suggest the manuscript to be accepted for publication.

Reviewer #2 (Remarks to the Author):

The authors addressed the points from the referee reports and I therefore advise publication of the manuscript. Thank you.